# Glycemic Index of Slowly Digestible Carbohydrate Alone and in Powdered Drink-Mix

**DOI:** 10.3390/nu11061228

**Published:** 2019-05-29

**Authors:** Vishnupriya Gourineni, Maria L. Stewart, Rob Skorge, Thomas Wolever

**Affiliations:** 1Global Nutrition R & D, Ingredion Incorporated, 10 Finderne Ave, Bridgewater, NJ 08807, USA; maria.stewart@ingredion.com (M.L.S.); rob.skorge@ingredion.com (R.S.); 2Glycemic Index Laboratories, Inc., 20 Victoria street, Toronto, ON M5 2N8, Canada; twolever@gilabs.com

**Keywords:** slowly digestible carbohydrate, glycemic index, steady glucose release

## Abstract

Consumer interest in food and beverages with carbohydrates offering steady glucose release and lower glycemic index (GI) continues to rise. Glycemic index is one of the metrics for carbohydrate quality. Slowly digestible carbohydrates (SDC) offer an ingredient solution to improve carbohydrate quality and meet consumer needs. SUSTRA^TM^ 2434 slowly digestible carbohydrate is a blend of tapioca flour and corn starch. The study objective was to determine the glycemic index of the SDC ingredient alone and in a powdered drink-mix. In a randomized, single-blind study, heathy adults (*n* = 14) consumed four test drinks, delivering 50 g available carbohydrates on separate days to measure GI. Participants either consumed dextrose in water (placebo), SDC ingredient in water, SDC drink-mix powder reconstituted in skim milk, or control drink-mix reconstituted in skim milk (without SDC). Post-prandial glucose response was measured over 4 h. SDC exhibited lower GI (0–2 h) and higher steady glucose release (beyond 2 h). SDC alone (GI = 27) and SDC in drink-mix (GI = 30.3) showed significantly lower GI (−27%) compared to dextrose (100) and the control drink-mix (41.5). SUSTRA^TM^ 2434 SDC is a low glycemic ingredient, suitable for product innovations with potential for low glycemic and steady glucose release claims.

## 1. Introduction

Carbohydrates are commonly consumed macronutrients, which serve as the primary energy source to cells. Food sources of carbohydrates also contribute fiber and other nutrients in the diet. Thus, the Institute of Medicine established an acceptable macronutrient distribution range (AMDR) for carbohydrates as 45–60% of total calories and a recommended daily allowance (RDA) as 130 g/d for adults and children aged >1 year old [1]. Structurally, carbohydrates are classified as mono and di-saccharides or sugars, starch, and non-starch polysaccharides. Carbohydrates are nutritionally categorized as either digestible/available or non-digestible; of the nondigestible carbohydrates, some may be fermentable [2]. Consumption of most glucose-containing sugars and rapidly digestible starches raise blood glucose and insulin levels, thereby increasing the risk for chronic conditions such as diabetes. In contrast, slowly digestible starches are digested gradually, releasing glucose steadily into the blood stream, due to structural characteristics or food matrices that result in prolonged enzymatic hydrolysis [3].

Carbohydrate digestion, absorption and its impact on post-prandial glucose response is conceptualized as glycemic index (GI). Carbohydrates are ranked as low GI (≤55), Medium GI (56–69), and High GI (≥70), indicating the extent to which available carbohydrate in a food raises blood glucose relative to equal weight of glucose. Glycemic index is considered as one of the metrics for carbohydrate quality [4]. Meta-analysis studies have confirmed low-GI diets to be associated with reduced risk of chronic diseases, such as diabetes [5]. Slowly digestible carbohydrates may reduce risk of chronic disease because they elicit a lower glycemic response and potentially stimulate intestinal hormones, increase satiety, and reduce food intake [6]. Foods with a high slowly digestible carbohydrate (SDC) content have low GI and are of greater relevance for improving public health, as substantiated by growing scientific evidence and regulatory claims [7].

Consumer perception on sugars has changed over the years as a result of public health messages related to its association with a global increase in obesity and diabetes. The recommendations of the World Health Organization (WHO) to reduce sugar intake to 10% of total energy and its adoption by various nations have also contributed to the change in consumer preferences [8]. In its 2015 dietary guidelines, the United States Department of Agriculture (USDA) highlighted the importance of carbohydrate quality by recommending consumers choose healthy food and beverages and limit calories from “added sugars” [9]. To improve the availability and easy identification of low-GI products, Australia introduced the “GI symbol” as a carbohydrate indicator on front-of-pack labelling and regulates the GI claims [10]. Market research indicated that over 80% of Australian consumers perceive the “GI symbol” on food products as a “wholesome choice, scientifically tested and provide sustained energy/glucose release” [10]. Sustained/steady glucose release may be interpreted as exerting steadier blood glucose rise, slowing glucose drop, and remaining above the baseline for a longer duration than control food or beverages.

Earlier research was focused on carbohydrate quantity, such as amount and types (sugars vs. starches, simple vs. complex carbs), as a dietary intervention strategy for type 2 diabetes. However, epidemiological and experimental studies demonstrated that both quantity and quality are important for improving diet quality [11,12]. Thus, products delivering carbohydrate benefits, such as calories or slow energy release (e.g., SDC), low-GI, nutritionally dense, and offering desirable sensory attributes, are preferred by healthy and diabetic populations [13]. Additionally, products with SDC and low-GI are desirable for athletes due to sustained release of systemic glucose [14]. However, these nutritional products, targeted at active and sedentary consumers, require consumer-friendly formulations, including nutritional bars, drink-mixes, etc. A few commercial products with high SDC are marketed for niche consumer segments targeted for diabetics [15].

A previous study evaluated in vitro and in vivo digestibility of SUSTRA^TM^ 2434 slowly digestible carbohydrate [16]. The study also assessed glycemic index of SDC in non-thermal applications, such as in a cold-pressed bar and a pudding. However, the measured glycemic index of an ingredient may vary in different food matrices. Therefore, the current study aimed to determine the glycemic index of SDC alone and in food form (powdered drink-mix). A powered drink-mix formulation was used for SDC inclusion due to its perceived benefit of convenience (reconstitution in different beverages) and in alignment with market trends. The secondary objective was to evaluate the glycemic response of the SDC drink-mix compared to a control drink-mix over four hours in a healthy population.

## 2. Materials and Methods

This study was conducted in accordance with the ethical principles outlined in the Declaration of Helsinki and the protocol was approved by the Western Institutional Review Board (Vancouver, BC, Canada). All subjects provided written informed consent prior to starting the study. The clinical study was conducted at GI Labs (Toronto, ON, Canada).

### 2.1. Subject Screening

Inclusion criteria: Participants were healthy males or non-pregnant females, 18–75 years of age, and with a body mass index (BMI) of ≥20 and ≤40 kg/m^2^. Participants were required to maintain their regular diet, supplement intake, physical activity and body weight throughout the study duration and refrain from smoking prior to each visit. On test days, subjects were not allowed to take any dietary supplements until dismissal from the GI labs. Subjects were required to have normal fasting serum glucose (<7.0 mmol/L capillary corresponding to whole blood glucose <6.3 mmol/L), abstain from alcohol consumption and avoid vigorous physical activity for 24 h prior to all test visits. Subjects had to understand the study procedures and be willing to provide informed consent to participate in the study and authorization to release relevant protected health information to the investigator.

Exclusion criteria: Subjects were excluded if they failed to meet inclusion criteria, had a history of chronic disease, such as type 1 or 2 diabetes, cardiovascular disease, cancer, gastrointestinal disorders, used medications within four weeks of the screening, had surgery within 3 months of screening, had an intolerance or allergy to test ingredients, had extreme dietary habits, had drastic body weight changes (>3.5 kg within four weeks of screening duration), had the presence of any symptoms of an active infection during screening or study visits, had a history of alcohol or substance abuse, or had an unwillingness or inability to comply with the experimental procedures and to follow GI Labs safety guidelines.

### 2.2. Study Design and Subjects

The study was a randomized, single-blinded, placebo-controlled, cross-over design, with 14 healthy adults. The order of the reference (50 g dextrose) and test foods was randomly assigned among 3 blocks as follows: block 1 consisted of one of the test foods and 50 g dextrose in random order, block 2 consisted of one of the test foods, and block 3 consisted of one of the test beverages and dextrose in random order. Randomization was performed using the RAND function on Excel 2010 (Microsoft Corp., Redmond, WA, USA). Orders were assigned to subjects in the order they attended for the first visit. Eligible participants were studied on five separate days over a period of 2 to 5 weeks. The interval between successive tests was no less than 48 h and no more than 2 weeks. Subjects completed five study visits in a random order, during which they consumed one of the following treatments: dextrose in water (placebo), SDC ingredient in water, SDC drink-mix powder reconstituted in skim milk, or control drink-mix reconstituted in skim milk (without SDC). The dextrose beverage was administered twice for glycemic index calculations. All participants (men = 9, women = 5) completed the study.

### 2.3. Study Foods

All test beverages contained 50 g available carbohydrates and were packaged in separate sachets. Dextrose (54.6 g) (Clintose^®®^ dextrose, Archer Daniels Midland Company, Decatur, IL, USA) and SDC ingredient (58.8 g) (SUSTRA^TM^ 2434 slowly digestible carbohydrate, Ingredion Incorporated, Bridgewater, NJ, USA) were mixed in 415.0 g and 411.8 g of water, respectively. Control drink-mix (46.2 g) was reconstituted in 426.5 g skim milk (Control Drink) and SDC drink-mix (48.2 g) was reconstituted in 444.6 g skim milk (SDC Drink). The reference food, 54.6 g dextrose, was tested twice by each subject. The composition of the beverages is shown in Table 1 and recipe/formulation is provided as a Appendix A. Each beverage was served with a drink of 1 or 2 cups of coffee, tea or water with 30 mL 2% milk, if desired. At the first visit, each subject selected the type and volume of drink desired; the same type and volume was consumed on all subsequent visits.

### 2.4. Study Visit Procedures

Participants were asked to maintain stable dietary and activity habits throughout the study. Prior to each study visit, participants refrained from drinking alcohol and from unusual levels of food intake or physical activity for 24 h. On each test occasion, subjects arrived at the clinical site after fasting for 10 to 12 h. Two fasting blood samples for glucose analysis (2–3 drops into a fluoro-oxalate tube) were obtained by fingerprick. 5 min apart and after the second sample, the subject started to consume a test beverage. Each beverage was served with a drink of 1 or 2 cups of coffee or tea with 30 mL of 2% milk, if desired, or water. At the first visit, each subject selected the type and volume of drink desired and the same type and volume of drink was consumed on subsequent visits. Subjects consumed the entire beverage within 10 min. At the first sip, a timer was started and additional blood samples for glucose analysis (2–3 drops into a fluoro-oxalate tube) were taken at 15, 30, 45, 60, 90, 120, 150, 180, 210 and 240 min after starting to eat. Blood samples were obtained from hands warmed with an electric heating pad for 3–5 min prior to each sample.

### 2.5. Biochemical Analysis

After blood collection the tubes containing blood for glucose analysis were rotated to mix the blood with an anti-coagulant and then placed in a refrigerator until the last blood sample in the set had been collected. After all tubes were collected from one subject, the tubes were stored in a −20 °C freezer until analysis. Analysis was performed within 3 days of the study visit, using a YSI model 2300 STAT analyzer (Yellow Springs, OH, USA).

### 2.6. Data Analysis and Statistics

With *n* = 14, there is 80% power to detect a difference in GI of 28–33%. This was considered adequate to the size of effects which may be detected for SDC vs. dextrose. Glycemic index values were calculated based on previously published methods [17]. The incremental area under the blood glucose response curves (iAUC), ignoring area below fasting and net incremental area under the curve (net iAUC), where values below the baseline were treated as negative values, were calculated using the trapezoidal rule. Paired t-tests were conducted on blood glucose values at individual time points, incremental area-under-curve (iAUC), and glycemic index using GraphPad Prism 7 (v 7.03, GraphPad Software, Inc., La Jolla, CA, USA). *p*-values < 0.05 were deemed statistically significant.

## 3. Results

Study demographics are shown in Table 2. All participants were healthy and completed the study. No adverse events and no protocol deviations were reported.

The SDC ingredient alone in water yielded a significantly lower glycemic index compared to dextrose when matched for 50 g available carbohydrate (Table 3). The SDC ingredient lowered blood glucose mean concentration and reduced iAUC (0–2 h) by 74% as compared to dextrose (Figure 1 and Table 3). Mean blood glucose concentrations for SDC were significantly lower at 15, 30, 45 and 60 min compared to dextrose, which is reflected in a smaller peak rise for SDC (Table 3). However, mean blood glucose concentrations for SDC were significantly higher at 120, 150, 180, 210 and 240 min resulting in higher netAUC (3–4 h) compared to dextrose.

Consumption of SDC drink-mix powder reconstituted in skim milk significantly lowered glycemic index by 27% compared to the control drink-mix (Table 4). Mean blood glucose concentrations for SDC drink-mix were significantly lower and its iAUC (0–2 h) reduced by 32% compared to the control drink-mix added to skim milk (Figure 2 and Table 4). Mean blood glucose concentrations for SDC drink-mix were significantly lower at 30, 45 and 60 min compared to the control drink-mix, resulting in smaller peak rise for SDC drink-mix. While, netAUC (3–4 h) for SDC drink-mix was less negative than the control drink-mix, avoiding hypoglycemia.

## 4. Discussion

The current study determined the glycemic index of the SDC ingredient and SDC incorporated in a powder beverage-mix in healthy adults. Results show lower GI (0–2 h) and steady energy release (beyond 2 h) in response to SDC alone and in food-form compared to control beverages. Current study findings with SDC in powder drink-mix are comparable to previous interventions of SDC in a cold-pressed nutritional bar and pudding [15]. The control foods (bar and pudding) were high GI formulations in the previous study, in contrast to the current study where control powder-mix is a low GI formulation. Glycemic response was amplified with high GI control bars and pudding. However, even with low GI control drink-mix, there is a significant difference in glycemic response between the two drink mixes, indicating the superior carbohydrate quality of SDC. This is not only evident from the lower glycemic response of SDC drink-mix within the first 2 h, but also its prolonged digestion reflected as a glycemic peak closer to the baseline for longer duration.

Although consumer trends are shifting away from “added sugars”, carbohydrates still remain as the primary energy source for the majority of the population. Products with acceptable sensory attributes (taste and texture), convenience, and nutritionally complete with low GI and steady glucose release are some of the requirements for active and sedentary populations. This market need is partially fulfilled by commercial products claiming low GI and steady glucose release for diabetics as the target population [18]. A meta-analysis and systemic review of randomized studies conducted in the past decade demonstrated positive effects of low GI diets on fasting blood glucose (short-term biomarker of glucose metabolism) and glycated hemoglobin (HbA1c) (long-term biomarker of glucose metabolism) in type 2 diabetic patients [19]. Mounting scientific evidence from epidemiological and dietary intervention studies indicated the importance of carbohydrate quantity and quality on reducing risk of chronic diseases. Thus, high-carbohydrate (dietary fiber) and low-fat diets are considered beneficial for weight management and well-being [20,21]. The International Carbohydrate Quality Consortium panel recognized low GI, slow carbohydrate digestion and absorption as effective approaches in reducing post-prandial glycemic response, a beneficial physiological benefit in glucose metabolism [10].

Glucose homeostasis is a normal physiological phenomenon. In a healthy population, blood glucose concentration rises, followed by digestion, and then strives back to baseline levels with the help of insulin to facilitate cellular uptake. In this study, the glycemic response of SDC is lowered compared to high GI dextrose, making it flatter and stay above the baseline levels, leading to less glucose perturbations. The blood glucose values after consuming SDC remained above baseline concentration for over 4 h, which indicates greater homeostatic balance. Better homeostatic control exerts lower stress on key organs, such as the pancreas and liver. This feature highlights the superior quality of SDC in improving glycemic health and may indirectly impact other metabolic factors through gut-hormones [22].

Lower glycemic response and steady glucose release are health benefits not only in diseased populations but also for healthy populations. SDC also have relevance in sports nutrition as they offer low GI and slow energy release solutions to athletes in preparation for endurance training [23]. Glycemic and insulinemic response of sugar-based SDC in a diabetic-enteric formula was positive compared to control formulas [24]. Despite the presence of sugar-based SDC, available in sports nutrition, starch-based SDC offer product functionality and nutritional benefits as demonstrated by its wide-range of product formulations.

The glycemic and steady glucose release benefits of slowly digestible carbohydrates may expand from diabetic and sedentary populations to healthy and active populations. Low GI and slowly digestible carbohydrates fed to pregnant insulin-resistant rats resulted in lower adipogenesis in their offspring compared to rapidly-digestible carbohydrates [25]. In another rodent study, consumption of SDC added to a high-fat diet for eleven weeks showed lower food intake, which was associated with suppression of appetite-stimulating hormones through the gut-brain axis [26]. These studies provide insights on the SDC benefits and its underlying mechanisms. In a clinical study [27], breakfast with high-SDC and low GI containing cereal products consumed for five weeks reduced appearance of glucose in the early part of the morning and extended the glucose release into the later part of the morning. The strength of the current study is a preliminary understanding of glycemic index and the steady glucose release effects of starch-based SDC alone and in food form of broader consumer relevance. The limitation of the study is the absence of insulin measurements.

## 5. Conclusions

Carbohydrate quantity and quality have a major influence on the risk of chronic diseases. Public health concerns, such as diabetes and obesity, have created a need for technological food innovations to improve diet quality. Low GI and slowly digestible carbohydrates offer solutions for both diseased and healthy populations. SUSTRA^TM^ 2434 slowly digestible carbohydrate, is a gluten-free ingredient that can be formulated into non-thermal applications, such as beverage mix and cold-pressed bars.

SUSTRA^TM^ 2434, a starch-based SDC, is a low GI ingredient providing steady glucose release benefits. Inclusion of SDC in powder drink-mix, a convenient food form, is shown to provide low GI and slow energy benefits. The study corroborates the functionality and nutritional benefits of starch-based SDC for broader populations and markets.

## Figures and Tables

**Figure 1 nutrients-11-01228-f001:**
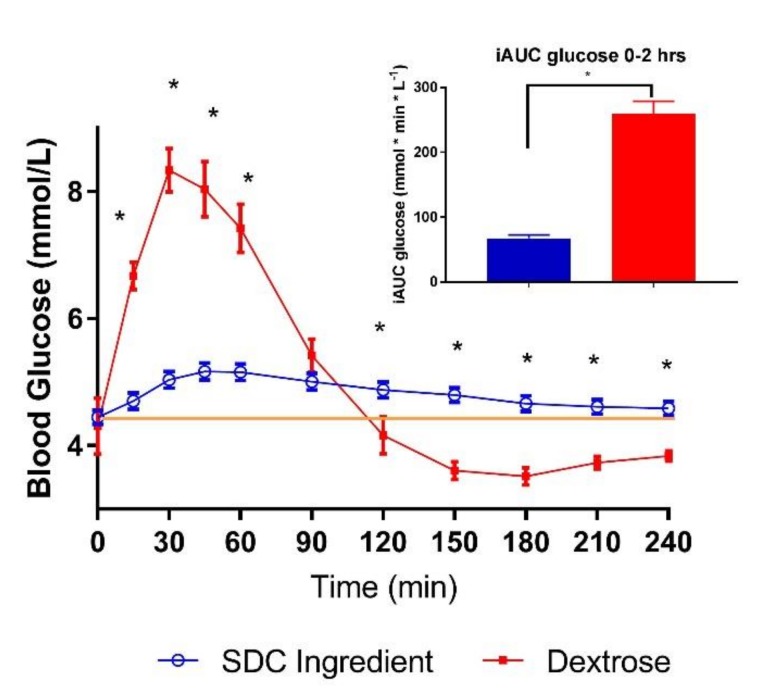
Post-prandial glycemic response of SDC and dextrose in healthy adults (*n* = 14). Data are mean ± standard error mean (SEM); Yellow line indicates baseline value. * indicates treatments were significantly different at specific time points in the paired *t*-test (*p* < 0.05).

**Figure 2 nutrients-11-01228-f002:**
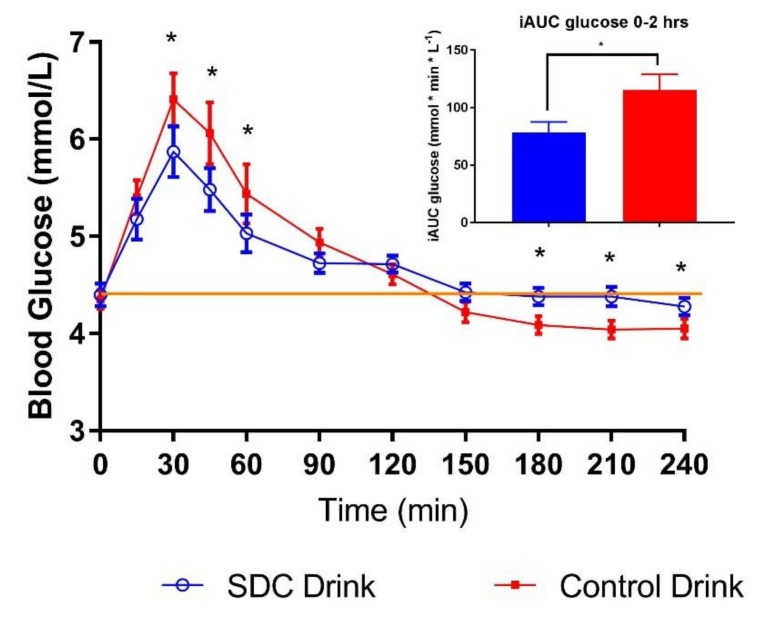
Post-prandial glycemic response of drink-mixes (SDC and Control) in healthy adults. Data are mean ± standard error mean (SEM); Yellow line indicates baseline value * indicates treatments were significantly different at specific time points in the paired *t*-test (*p* < 0.05).

**Table 1 nutrients-11-01228-t001:** Nutrient composition of slowly digestible carbohydrates (SDC) alone and in powdered drink-mix.

Nutrient Content (g)	Dextrose	SDC	SDC Drink-Mix	Control Drink-Mix
Serving size	469.6	473.2	492.8	472.7
Total carbohydrates	50.0	52.0	51.0	50.0
Available carbohydrates	50.0	50.0	50.0	50.0
Sugars	50	0.0	29	36
Dietary fiber	0.0	2.0	1.0	0.0
Protein	0.0	0.0	22.0	21.0
Fat	0.0	0.0	3.0	3.0

Dextrose and SDC were mixed in water. Drink-mixes (SDC and control) were reconstituted in skim milk.

**Table 2 nutrients-11-01228-t002:** Demographics of study participants.

Participants (*n* = 14)	Mean ± Standard Deviation (SD)
Age (years)	46.6 ± 14.4
Gender (male/female)	9/5
Weight (kg)	78.9 ± 19.4
Body mass index (kg/m^2^)	27.3 ± 5.2
Fasting blood glucose (mmol/L)	4.4 ± 0.5

**Table 3 nutrients-11-01228-t003:** Post-prandial glycemic response of SDC and dextrose.

Outcomes	SDC	Dextrose
Glycemic index *	27.0 ± 1.9	100
iAUC (0–2 h) *	67.3 ± 5.0	259.3 ±19.4
netAUC (3–4 h) *	12.8 ± 4.6	−36.3 ± 7.6
Peak rise (mmol/L) *	0.94 ± 0.04	4.29 ± 0.32

Glycemic Index (GI) Scale: Low GI: ≤55; Medium GI: 56–69; High GI: ≥70 * Values are presented as mean ± standard error mean (SEM), dextrose (control) vs. SDC statistically different by paired *t*-test (*p* < 0.05). iAUC = Incremental area-under-curve, netAUC = Net incremental area-under-curve (mmol × min/L).

**Table 4 nutrients-11-01228-t004:** Post-prandial glycemic response of drink-mixes.

Outcomes	SDC Drink-Mix	Control Drink-Mix
Glycemic index *	30.3 ± 2.7	41.5 ± 3.4
iAUC (0–2 h) *	78.6 ± 9.0	115.3 ± 13.9
netAUC (3–4 h) *	−4.2 ± 4.8	−20.9 ± 7.6
Peak rise (mmol/L) *	1.52 ± 0.17	2.2 ± 0.2

Glycemic Index (GI) Scale: Low GI: ≤55; Medium GI: 56–69; High GI: ≥70 * Values are presented as mean ± standard error mean (SEM), control drink-mix vs. SDC drink-mix statistically different by Paired t-test (*p* < 0.05, netAUC = Net incremental area-under-curve (mmol × min/L)). iAUC = Incremental area-under-curve

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
