# Peer review of "Glycemic Index of Slowly Digestible Carbohydrate Alone and in Powdered Drink-Mix"

_nutrients, 2019, doi:10.3390/nu11061228_

Round 1

Reviewer 1 Report

The article of Gourineni et al reports GI measures of a slow-digested carbohydrate administered in the form of a beverage. While the scope of the research is of interest (comparing GI of ingredient alone vs in food product) the manuscript lacks precision and many key concepts are not sufficiently described or discussed.

Abstract :

l. 20 /23 « sustained energy » is not a scientifically well definedd concept. Please use an alternative term in the abstract (e.g. glucose response 3-4h), unless clear description is provided

l. 20. Please provide GI values in abstract, and % reduction compared to controls (both alone and in beverage)

INTRO

General comments: Some concepts should be explained in the introduction:

- SDC: more details are needed on the ingredient. Are there natural sources? (pulses, raw bananas, …)? What are the characteristics of SUSTRA (characterization of linkages, molecular weight, production process, original food source, molecular weight, previous studies on digestibility…?)

- Comparison of food matrix: how is the beverage processing method different from cold-bar / pudding process? How can these modify the properties of SUSTRA and justify a GI measure in a different food matrix?

- “sustained energy”: authors should describe and provide references on what they mean by “sustained energy”: higher blood glucose response after 3-4 hours? Slow glucose release? More energy for physical activity? This is not generally recognized as a health benefits, hence the importance to clarify the meaning.  

Others:

l. 28 why energy only to brain and red blood cells?? Glucose is main energy source of almost all cell types

l. 63. It is confusing to say that carbohydrate quantity should be high without putting into context, especially when dietary guidelines recommend to decrease energy intake and density in sedentary populations. This sentence should be rephrased

METHODS:

Some sections are confusing, it would be good to clarify and be consistent in the method description:

l. 122: what was the control drink made of? Please provide recipe/ingredient list

l. 124: are all test meals beverages? If so, please replace the word “test meal” by “test beverages”

Why were the beverages served with coffee/tea/milk? Both coffein and milk are expected to affect GI, but this is not discussed later in results + discussion. Please clarify the rationale

l. 139: why is there a “bite” if test meals were beverages?

Table 3: why is the GI reference for dextrose, while it was mentioned in methods line 105 that glucose was used as control? If dextrose GI value is 100, than it would be good to add SEMs

Table 3: why is there a need for alphabetic letters from a to d, when only paired-comparison were made? (presumably between SDC and control conditions)

l. 185. Please delete “avoiding energy crash”. This can be mentioned in the discussion, but then what is meant by “energy crash” must be explained and referenced

l. 185. “drink mix was higher than control” please rephrase by “drink mix was less negative than control”

DISCUSSION

Overall, the discussion doesn’t describe the main findings and does not reflect initial objective (measure of GI in beverage). I suggest to include in the discussion the following aspects:

Comparison of results between beverages and other food forms (cold press bars/puddings). Possibly use other slow-digestible ingredients that have been compared in different food forms for glucose response

Comparison of GI of SDC alone or in beverage (with milk): what is the milk/beverage form adding, and how can the difference in GI reduction be explained

l. 216: why is glucose response attenuated in beverages? One would expect a higher glucose response in liquids due to faster gastric emptying. Was the carbohydrate quantity different in previous studies? Please explain and provide hypotheses.

l. 230. “sustained energy” : c.f. general comment above

l. 234. “positive” in reference to what?

l. 234-236: “although sugar-based SDC are available (…)”. This sentence is not clear, please consider rephrasing or delete.

Paragraph 237-245: the discussion on detailed regulation mechanisms and animal models seems out of place and it is difficult to make a link with the rest of the discussion. It can be relevant to discuss potential mechanisms of action, but they should be better integrated in the rest of the discussion, which is mainly focused on basic markers measured in humans.

L. 246-247. I disagree that the study provides understanding of GI and sustained energy effects of starch-based SDC alone and in food form. To be the case, it would be worth to consider adding some discussion about potential mechanisms of action (due to either SDC or food form) and link between technical properties of the ingredient/food product and physiological response observed. Some mechanisms explaining the lower GI should be better described. How can it be ensured that the full amount of SDC was digested/absorbed (is there any previous data to provide as reference)?

L. 248: how can it be ensured that lower GI is not due to higher insulin secretion? Is there any previous data justifiying this?

Author Response

·       The article of Gourineni et al reports GI measures of a slow-digested carbohydrate administered in the form of a beverage. While the scope of the research is of interest (comparing GI of ingredient alone vs in food product) the manuscript lacks precision and many key concepts are not sufficiently described or discussed.

·       Abstract :

·       l. 20 /23 « sustained energy » is not a scientifically well defined concept. Please use an alternative term in the abstract (e.g. glucose response 3-4h), unless clear description is provided

·       Response: Revised the term “sustained energy” to “steady glucose release” throughout the manuscript

·       l. 20. Please provide GI values in abstract, and % reduction compared to controls (both alone and in beverage)

·       Response: Added GI values as suggested.

·       INTRO

·       General comments: Some concepts should be explained in the introduction:

·       - SDC: more details are needed on the ingredient. Are there natural sources? (pulses, raw bananas, …)? What are the characteristics of SUSTRA (characterization of linkages, molecular weight, production process, original food source, molecular weight, previous studies on digestibility…?)

·       Response: SUSTRA™ SDC is a physical dry blend of corn starch and Tapioca with > 40% of SDC fraction as measured by Englyst assay. The ingredient processing is trade protected. A 2017 published study summarized the research findings on ingredient digestibility (in-vitro and in-vivo) and also reported post-prandial glycemic response of SDS incorporated in a cold-pressed bar and pudding.  (Earlier publication: Gourineni, V.; Stewart, M.L.; Skorge, R.; Sekula, B.C. Slowly Digestible Carbohydrate for Balanced Energy: In Vitro and In Vivo) Evidence. Nutrients 20179, 1230.

·       - Comparison of food matrix: how is the beverage processing method different from cold-bar / pudding process? How can these modify the properties of SUSTRA and justify a GI measure in a different food matrix?

·       Response: SDC performance is limited to non-thermal food and beverage applications. Processing native cereal starches (inherently SDS/RS materials) in presence of heat and moisture diminishes SDS/RS content and increases RDS content due to the disorganization of the starch structure called starch gelatinization and thus increases enzymatic susceptibility during digestion and rapid release of glucose. This study (powder drink-mix) and earlier published study utilized SDC incorporated into non-thermal applications (cold-pressed bar, pudding). The processing differences did not impact SDC content in the test foods. The Differences in GI of SDC in cold-pressed bars, pudding and powder drink mix could be due to formulation differences as noted in the discussion section lines 211 - 218

·       - “sustained energy”: authors should describe and provide references on what they mean by “sustained energy”: higher blood glucose response after 3-4 hours? Slow glucose release? More energy for physical activity? This is not generally recognized as a health benefits, hence the importance to clarify the meaning.  

·       Response: ‘Sustained energy” is not scientifically defined or well-regulated term. However, “sustained energy” claim is seen in some of the marketed food and supplements categories. In a recent publication by Marinangeli C.P.F. & Harding S.V. ((2016), J Hum Nutr Diet. 29, 401–404), the author suggest that the term can have multiple consumer perceptions. The literature is limited in this area. Sustained glucose release may be interpreted as to exert steadier blood glucose rise, slow glucose drop and remaining above the baseline for longer duration than control food or beverage. Revised by including “steady glucose release” and removed “sustained energy”.

·       Others:

·       l. 28 why energy only to brain and red blood cells?? Glucose is main energy source of almost all cell types

·       Response: Yes. Deleted “brain and red blood cells”.

·       l. 63. It is confusing to say that carbohydrate quantity should be high without putting into context, especially when dietary guidelines recommend to decrease energy intake and density in sedentary populations. This sentence should be rephrased

·        Response: Revised the statements to avoid confusion and increase clarity to the readers

·       METHODS:

·       Some sections are confusing, it would be good to clarify and be consistent in the method description:

·       l. 122: what was the control drink made of? Please provide recipe/ingredient list

·       Response: SDC ingredient mixed in water is compared with dextrose control mixed in water. Powder drink-mixes, control and SDC incorporated, were mixed in skim milk. This is clearly mentioned in section # 2.3 and nutrient composition of study beverages were listed in Table 1. The recipe is similar to regular powder drink-mixes with maltodextrin. The test drink-mix replaces some fraction of granulated sugar and maltodextrin with SDC/SDS ingredient. The drink-mix recipe is provided as a supplementary table.

·       l. 124: are all test meals beverages? If so, please replace the word “test meal” by “test beverages”

·       Response: Revised as beverage in the entire manuscript

·       Why were the beverages served with coffee/tea/milk? Both coffee and milk are expected to affect GI, but this is not discussed later in results + discussion. Please clarify the rationale

·       Response: Certainly, a drink of coffee/tea has a small effect on the glycemic response - but the same drink is given with each test meal (including the reference of glucose), and thus the effect on the glycemic response is controlled for.  A drink is given with the test meals to ensure the volume of the meal is large enough to stimulate gastric emptying.  There are 2 reasons why we allow subjects to choose to drink coffee or tea with each test meal (or they may have water if they wish, but most choose coffee/tea).  1. many subjects are used to having coffee or tea with breakfast and being able to have this with the test meals makes the experience more normal and pleasant for them; this may make the results more physiologically relevant than if the test meal was unpalatable and unpleasant.  2. Earlier research showed (Wolever, Effect of coffee and tea on the glycaemic index of foods: no effect on mean but reduced variability British Journal of Nutrition (2009), 101, 1282–1285) when measuring GI values, compared to drinking only water, having coffee/tea with the test meals does not affect the mean GI, but reduces the SD by about 30%, an effect which improves statistical power (attached).  The presumed reason for this is that coffee/tea stimulate gastric emptying and, therefore, provide a more consistent glycemic response from day-to-day.

·       l. 139: why is there a “bite” if test meals were beverages?

·       Response: Deleted, Changed to  “sip

·       Table 3: why is the GI reference for dextrose, while it was mentioned in methods line 105 that glucose was used as control? If dextrose GI value is 100, than it would be good to add SEMs

·       Response: Changed “50 g glucose” to “50 g dextrose” in the manuscript

·       Table 3: why is there a need for alphabetic letters from a to d, when only paired-comparison were made? (presumably between SDC and control conditions)

·       Response: Corrected

·       l. 185. Please delete “avoiding energy crash”. This can be mentioned in the discussion, but then what is meant by “energy crash” must be explained and referenced

·       Response: Revised to “hypoglycemia”

·       l. 185. “drink mix was higher than control” please rephrase by “drink mix was less negative than control”

·        Response: Revised as suggested

·       DISCUSSION

·       Overall, the discussion doesn’t describe the main findings and does not reflect initial objective (measure of GI in beverage). I suggest to include in the discussion the following aspects:

·       Comparison of results between beverages and other food forms (cold press bars/puddings). Possibly use other slow-digestible ingredients that have been compared in different food forms for glucose response

·       Response: The GI of SDC in other food form (cold-pressed bar/puddings) are published in 2017. The comparison is included in the discussion section

·       Comparison of GI of SDC alone or in beverage (with milk): what is the milk/beverage form adding, and how can the difference in GI reduction be explained

·       Response: we intend to show GI of SDC ingredient. So, it has to be compared with a true control. For this study, we also chose to incorporate SDC ingredient in food/beverage application (powder drink-mix) to evaluate its GI comparing with a control powder-mix.

·       l. 216: why is glucose response attenuated in beverages? One would expect a higher glucose response in liquids due to faster gastric emptying. Was the carbohydrate quantity different in previous studies? Please explain and provide hypotheses.

·       Response: Attenuation of glucose response could be attributed to carbohydrate identity  e.g. lactose is only 50% glucose while maltodextrin is 100% glucose. In this study and previously published studies (SDC in bars and pudding), available glucose was matched for control and test meals/beverages.

·       l. 230. “sustained energy” : c.f. general comment above

·       Response: Added steady blood glucose and responded to all questions related to sustained energy

·       l. 234. “positive” in reference to what?

·       Response: Revised the sentence

·       l. 234-236: “although sugar-based SDC are available (…)”. This sentence is not clear, please consider rephrasing or delete.

·       Response: Revised the sentence

·        

·       Paragraph 237-245: the discussion on detailed regulation mechanisms and animal models seems out of place and it is difficult to make a link with the rest of the discussion. It can be relevant to discuss potential mechanisms of action, but they should be better integrated in the rest of the discussion, which is mainly focused on basic markers measured in humans.

·       Response: The animal studies were included as a supplementary evidence for SDC benefits in general. The SDC used in this study is different vs the SDC used in animal model studies. The animal studies provide insights on underlying mechanisms, but these mechanisms may differ for various SDC’s.

·        

·       L. 246-247. I disagree that the study provides understanding of GI and sustained energy effects of starch-based SDC alone and in food form. To be the case, it would be worth to consider adding some discussion about potential mechanisms of action (due to either SDC or food form) and link between technical properties of the ingredient/food product and physiological response observed. Some mechanisms explaining the lower GI should be better described. How can it be ensured that the full amount of SDC was digested/absorbed (is there any previous data to provide as reference)?

·       Response: The current study and previously published study (Nutrients 20179, 1230) emphasizes on SDC content of the ingredient. The lower GI and steady glucose release is due to slower digestion of the ingredient. This is digestible at a slower rate due to less starch gelatinization of starch particle which is shown in in-vitro and in-vivo digestibility results (Nutrients 20179, 1230)

·       L. 248: how can it be ensured that lower GI is not due to higher insulin secretion? Is there any previous data justifiying this?

·       Response: The limitation of the current study is lack of insulin measurement, which is clearly stated in the discussion. Without having measured insulin, we cannot prove that the lower GI is not due to higher insulin secretion.  However, the SDS test beverage  contains nothing but carbohydrate - there is no protein and fat.  Therefore there is nothing in it which would stimulate insulin secretion or increase postprandial insulin.  Consuming glucose slowly (Jenkins DJA, Wolever TMS, Ocana AM, Vuksan V, Cunnane SC, Jenkins M, Wong GS, Singer W, Bloom SR, Blendis LM, Josse RG.  Metabolic effects of reducing rate of glucose ingestion by single bolus versus continuous sipping.  Diabetes 1990:39:775-81.), or consuming a liquid formula diet slowly (Wolever TMS.  Metabolic effects of continuous feeding.  Metabolism 1990;39:947-51.), to mimic the effect of slow digestion, reduces both glucose and insulin responses.  In general, foods with a low GI also elicit low insulin responses; the only exception to this appears to be dairy products whose low GI might be due in part to their higher insulinemic index associated with their high content of protein (Wolever TMS.  Yogurt is a low-glycemic index food.  J Nutr 2017;147:1462S-67S).  In the present study, SDS alone contained no protein, and the same amount of protein was present in the control and SDS beverages.

Reviewer 2 Report

Although the paper was very well written and the methods and statistical analysis were sound, I do not feel there is much relevance to mainstream science.  The study revolves around one particular product that has a low GI and therefore is not very relevant.  It is already common knowledge among the nutrition community that low glycemic foods are a healthier choice.  This paper does not add much significance to the literature.

Additionally, I do sense a conflict of interest in that the paper was written by researchers employed by a company that manufacturers sweeteners. 

Author Response

Although the paper was very well written and the methods and statistical analysis were sound, I do not feel there is much relevance to mainstream science.  The study revolves around one particular product that has a low GI and therefore is not very relevant.  It is already common knowledge among the nutrition community that low glycemic foods are a healthier choice.  This paper does not add much significance to the literature.

Response: The clinical study was aimed to determine the glycemic index of a slowly digestible carbohydrate (SDC). The ingredient is a starch-based blend of Tapioca flour and corn starch. There is common knowledge on health benefits of low GI foods such as dietary fiber, pulse proteins etc. However, consumers and food industry are interested in carbohydrates with low GI and steady release of energy. The glycemic and steady glucose release  benefits of slowly digestible carbohydrates are applicable to broad populations - diabetic, sedentary, healthy and active. The study provides clinical evidence supporting SDS benefits and offers new SDC for foods formulations

Additionally, I do sense a conflict of interest in that the paper was written by researchers employed by a company that manufacturers sweeteners. 

Response: There is no conflict of interest as the study sponsor manufactures and markets wide range of carbohydrate and protein ingredients. The current study used dextrose (sweetener) in the control, which was replaced by slowly digestible carbohydrate in the test beverages.

Reviewer 3 Report

The reviewer thanks the authors for the interesting research study, which addresses the emerging issue of introducing alternative carbohydrate in food formulation with nutritional function. Please find the following comments.

Line 1: Should the word "Foods" be changed to "Beverages" since this study only investigated the drinks of different SDC content?

Line 39: The authors talked about the "Carbohydrate digestion". However, "carbohydrate absorption" is another important factor in postprandial glycaemic excursion. Please include.

Line 39: It would be good to introduce the definition of slowly digestible CHO and rapidly digestible CHO, and perhaps resistant starch? And how they affect the acute glycaemia.

Line 85: The inclusion criteria of BMI and age seem to include a very wide range. Is it possible to include some participants with metabolic disease or problem, such as pre-diabetes?

Line 90: Could it be possible to include some participants with pre-diabetes?

Line 104: Did the authors use blocked randomisation?

Line 114: Does the "control drink-mix" contain dextrose?

Line 128: Please specify the free sugar content in all the treatments.

Line 125: Would the coffee affect the postprandial glycaemic responses?

Line 161: Table 2. Please use the same unit for all the blood glucose.

Line 170: Table 3. Please add the unit for the Peak Rise. Is it the estimated Peak Rise?

Line 197: Reducing added sugar is not contradictory to maintaining total CHO consumption. In fact, we would encourage increasing the consumption of the whole grain food as CHO source to improve the dietary fibre consumption.

Line 197-211: It is recommended to summary the main findings of the study in the first paragraph of the discussion. The first and second paragraphs seem to be better fitted in the introduction.

Author Response

The reviewer thanks the authors for the interesting research study, which addresses the emerging issue of introducing alternative carbohydrate in food formulation with nutritional function. Please find the following comments.

Line 1: Should the word "Foods" be changed to "Beverages" since this study only investigated the drinks of different SDC content?

Response: The title is changed from “Foods” to “Powdered drink-mix” (line # 3).

Line 39: The authors talked about the "Carbohydrate digestion". However, "carbohydrate absorption" is another important factor in postprandial glycaemic excursion. Please include.

Response: Revised as “carbohydrate digestion and absorption”

Line 39: It would be good to introduce the definition of slowly digestible CHO and rapidly digestible CHO, and perhaps resistant starch? And how they affect the acute glycaemia.

Response: Starch digestion rate measurement using Englyst method classifies RDS, SDS and RS. A 2017 published paper on the same ingredient highlighted the importance of the starch digestion rate based on structure and composition of starch granules. The current manuscript also provided an overview of SDS/SDC in the introduction section.

Line 85: The inclusion criteria of BMI and age seem to include a very wide range. Is it possible to include some participants with metabolic disease or problem, such as pre-diabetes?

Response: The clinical study conducted in 2018 included only healthy participants. We will consider “pre-diabetic populations” for future studies.

Line 90: Could it be possible to include some participants with pre-diabetes?

Response: The clinical study conducted in 2018 included only healthy participants. We will consider “pre-diabetic populations” for future studies.

Line 104: Did the authors use blocked randomisation?

Response: Yes, all treatments are randomized. The order of the reference (50g dextrose ) and test foods was randomly assigned among 3 blocks as follows: block 1 consisted of one of the test foods and 50 g dextrose in random order, block 2 consisted of one of the test beverages  and block 3 consisted of one of the test beverages  and 50 g dextrose  in random order. Randomization was performed using the RAND() function on Excel 2010 (Microsoft Corp., Redmond, WA). Orders were assigned to subjects in the order they attend for the first visit.

Line 114: Does the "control drink-mix" contain dextrose?

Response: Yes, the control used is dextrose mixed in water, while “control drink-mix” recipe is provided as a supplementary table. as noted in section – 2.3 of the manuscript.

Line 128: Please specify the free sugar content in all the treatments.

Response: Added to the nutrient composition shown as Table 1

Line 125: Would the coffee affect the postprandial glycaemic response?

Response: Certainly, a drink of coffee/tea has a small effect on the glycemic response - but the same drink is given with each test meal (including the reference of glucose), and thus the effect on the glycemic response is controlled for.  A drink is given with the test meals to ensure the volume of the meal is large enough to stimulate gastric emptying.  There are 2 reasons why we allow subjects to choose to drink coffee or tea with each test meal (or they may have water if they wish, but most choose coffee/tea).  1. many subjects are used to having coffee or tea with breakfast and being able to have this with the test meals makes the experience more normal and pleasant for them; this may make the results more physiologically relevant than if the test meal was unpalatable and unpleasant.  2. Earlier research showed (Wolever, Effect of coffee and tea on the glycaemic index of foods: no effect on mean but reduced variability British Journal of Nutrition (2009), 101, 1282–1285) when measuring GI values, compared to drinking only water, having coffee/tea with the test meals does not affect the mean GI, but reduces the SD by about 30%, an effect which improves statistical power (attached).  The presumed reason for this is that coffee/tea stimulate gastric emptying and, therefore, provide a more consistent glycemic response from day-to-day

Line 161: Table 2. Please use the same unit for all the blood glucose.

Response: Changed the unit for fasting blood glucose on Table 2, for consistency

Line 170: Table 3. Please add the unit for the Peak Rise. Is it the estimated Peak Rise?

Response: Added units for peak rise on Tables 3 and 4

Line 197: Reducing added sugar is not contradictory to maintaining total CHO consumption. In fact, we would encourage increasing the consumption of the whole grain food as CHO source to improve the dietary fibre consumption.

Response: Agreed with reviewer’s comments. Line # 193 notes consumer trends and perceptions for “added sugars” and highlights the importance of carbohydrates encompassing functional and nutritional benefits.

Line 197-211: It is recommended to summary the main findings of the study in the first paragraph of the discussion. The first and second paragraphs seem to be better fitted in the introduction.

Response: Revised the discussion section

Round 2

Reviewer 2 Report

The authors have addressed the problems.